# Logistic Regression Algorithm Differentiates Gulf War Illness (GWI) Functional Magnetic Resonance Imaging (fMRI) Data from a Sedentary Control

**DOI:** 10.3390/brainsci10050319

**Published:** 2020-05-25

**Authors:** Destie Provenzano, Stuart D. Washington, Yuan J. Rao, Murray Loew, James N. Baraniuk

**Affiliations:** 1Georgetown University Medical Center, Washington, DC 20007, USA; sdw4@georgetown.edu; 2Department of Biomedical Engineering, The George Washington University School of Engineering and Applied Science, Washington, DC 20052, USA; loew@gwu.edu; 3Division of Radiation Oncology, The George Washington University School of Medicine and Health Sciences, Washington, DC 20052, USA; yrao@mfa.gwu.edu

**Keywords:** Functional Magnetic Resonance Imaging (fMRI), Gulf War Illness (GWI), logistic regression, machine learning

## Abstract

Gulf War Illness (GWI) is a debilitating condition characterized by dysfunction of cognition, pain, fatigue, sleep, and diverse somatic symptoms with no known underlying pathology. As such, uncovering objective biomarkers such as differential regions of activity within a Functional Magnetic Resonance Imaging (fMRI) scan is important to enhance validity of the criteria for diagnosis. Symptoms are exacerbated by mild activity, and exertional exhaustion is a key complaint amongst sufferers. We modeled this exertional exhaustion by having GWI (*n* = 80) and sedentary control (*n* = 31) subjects perform submaximal exercise stress tests on two consecutive days. Cognitive differences were assessed by comparing fMRI scans performed during 2-Back working memory tasks before and after the exercise. Machine learning algorithms were used to identify differences in brain activation patterns between the two groups on Day 1 (before exercise) and Day 2 (after exercise). The numbers of voxels with *t* > 3.17 (corresponding to *p* < 0.001 uncorrected) were determined for brain regions defined by the Automated Anatomical Labeling (AAL) atlas. Data were divided 70:30 into training and test sets. Recursive feature selection identified twenty-nine regions of interest (ROIs) that significantly distinguished GWI from control on Day 1 and 28 ROIs on Day 2. Ten regions were present in both models between the two days, including right anterior insula, orbital frontal cortex, thalamus, bilateral temporal poles, and left supramarginal gyrus and cerebellar Crus 1. The models had 70% accuracy before exercise on Day 1 and 85% accuracy after exercise on Day 2, indicating the logistic regression model significantly differentiated subjects with GWI from the sedentary control group. Exercise caused changes in these patterns that may indicate the cognitive differences caused by exertional exhaustion. A second set of predictive models was able to classify previously identified GWI exercise subgroups START, STOPP, and POTS for both Days 1 and Days 2 with 67% and 69% accuracy respectively. This study was the first of its kind to differentiate GWI and the three sub-phenotypes START, STOPP, and POTS from a sedentary control using a logistic regression estimation method.

## 1. Introduction

Gulf War Illness (GWI) is a debilitating condition characterized by chronic widespread pain, fatigue, and cognitive impairment that are worsened by mild to moderate exertion (post-exertional malaise or exertional exhaustion) [1]. GWI affects approximately 25–30% of the 700,000 individuals that served in the 1990–1991 Persian Gulf War [2,3]. Although military exposures to nerve agents, pyridostigmine bromide pills and other potential neurotoxins have been linked to neurological and other findings in epidemiological studies of veterans and animal models, there are no validated clinical biomarkers, and the underlying pathological mechanisms remain unknown [4]. 

Functional magnetic resonance imaging in recent literature has shown promise as a potential differentiator of GWI from a sedentary control by examinations of regions that are activated or deactivated at rest or during tasks, and changes in brain activation that were caused by exertion [5,6,7,8,9,10,11,12]. Our group previously found that a submaximal exercise stress test was able to uncover neurological differences in the fMRI data of the Gulf War Illness subgroups (START: Stress Test Activated Reversible Tachycardia) and (STOPP: Stress Test Originated Phantom Perception) while performing a test of attention (the N-back working memory task) [13]. 

Using this same protocol, we also demonstrated that Chronic Fatigue Syndrome (CFS) fMRI data was differentiable from a sedentary control using a multivariate pattern of activation and machine learning (a logistic regression algorithm) [14]. As a result, we hypothesized that a machine learning algorithm such as a logistic regression estimation method used on the fMRI data of GWI subjects after exercise would potentially identify multivariate patterns of brain activation to distinguish GWI and its sub-groups from control subjects. 

We modeled cognitive aspects of exertional exhaustion using a consecutive day submaximal bicycle exercise paradigm. Blood oxygenation level dependent (BOLD) signals were measured during an N-Back (2-Back > 0-Back) working memory task that was performed prior to and after the pair of stress tests [15]. BOLD activity from the cognitive tasks were mapped to Automated Anatomical Labeling Atlas (AAL) regions and fed as model inputs into a logistic regression [16]. Data were split into training and testing sets. Separate predictive models were constructed for pre- and post-exercise periods to evaluate changes in brain activation caused by exertion as found in a previous dual exercise study [13]. The outcome of the training set was a logistic regression classifier that defined a multi-region pattern of AAL regions to predict GWI versus control status. The model was cross-validated on the testing set to ensure performance. This study detailed that a machine learning algorithm used on fMRI data gathered from the N-Back working memory task before and after exercise was able to classify GWI from a sedentary control population.

## 2. Materials and Methods

### 2.1. Ethics

Subjects provided written informed consent both for participation and for use of all data in publications. Our study was approved by the Georgetown University Institutional Review Board, (IRB 2009-229, 2013-0943, 2015-0579), U.S. Army Medical Research and Material Command (USAMRC), the Human Research Protection Office (HRPO A-155547.0, A-18749), and registered on clinicaltrials.gov as NCT00810225, NCT01291758, NCT03560830. All of our clinical investigations were run in accordance with the principles of the Declaration of Helsinki.

### 2.2. Approach

Our system to generate a probabilistic predictive model from fMRI BOLD data was a logistic regression predictive model built on top of voxel data consolidated into anatomical regions of interest delineated by the Automated Anatomical Labeling (AAL) atlas [14]. The input variables of the model were numbers of significantly activated voxels per region. The model and grouping methodologies were appropriate for our setting as we evaluated the binary outcome variable of GWI vs. control status. The output was the map of AAL regions that collectively created a pattern to significantly distinguish GWI from control with high statistical significance. Similar studies have sought to evaluate fMRI data by seeking regions of significant differences in relative levels of BOLD activity between groups. As such, the use of a logistic regression provided a novel take.

The process followed a standard method for fMRI feature extraction, model build, validation, and performance evaluation (Figure 1) [17]. Features were extracted from pre-processed data, iteratively passed through a feature reduction method, trained on a set of learning data, tested on a validation set, and cross validated to evaluate performance. 

### 2.3. Subjects

Candidates responded to online and personal contacts and gave verbal informed consent for screening by telephone. After explaining the protocol and assessing chronic medical and psychiatric diseases, 105 of the 216 candidates declined to participate or were excluded from participation [1,3]. GWI was confirmed in person using the Chronic Multisymptom Illness (CMI) and Kansas criteria. The Kansas criteria require moderate or severe chronic symptoms in at least three of six domains: fatigue/sleep, muscle/joint pain, neurological/cognitive/mood, gastrointestinal, respiratory, and skin symptoms [1]. Complete history and physical examination, exercise, and fMRI data were collected for 80 GWI and 31 control subjects (*n* = 111). All subjects had a sedentary lifestyle with less than 40 min of active aerobic work or exercise per week. 

Further information regarding the study protocol, screening, demographics, subject symptoms, subject pain perception, orthostatic measurements, interoceptive complaints, chemical sensitivity questionnaires, and subject quality of life domain data are reported in previous published articles from our group and online as Appendix A [13,18,19,20]. It should be noted that all subjects were screened for ability to perform the task prior to fMRI data collection, and were able to practice the N-back memory task until they felt comfortable prior to recording. Performance data from the GWI group was reported in a previous study to be markedly lower (on average about 26% lower) than the sedentary control both before and after exercise [13]. 

We admitted subjects to the Georgetown Howard Universities Clinical Translation Science Clinical Research Unit as described previously [21]. After overnight rest, they first performed the zero-back and two-back portions of the N-back working memory tasks while undergoing an\fMRI scan, and then practiced the submaximal exercise stress test (Day 1). The next day they performed the same exercise test then repeated the fMRI scanning procedure and cognitive tests.

The continuous N-Back task assesses verbal working memory and attention [13,15]. Subjects practiced the task in a mock scanner until satisfactory performance was achieved. The task had five blocks of fixation, 0-Back and 2-Back components. Each block began with fixation as subjects viewed a blank screen for eight seconds. 0-Back testing proceeded by observation of strings of nine letters for two seconds each. The letters (A, B, C, D) were shown at random. Subjects viewed each letter and pressed a corresponding button with both hands on a fiber-optic button box (ePrime software) [22]. After a second fixation period, an additional string of nine letters was shown for the 2-Back task. Subjects were shown a partial string of letters where they were encouraged to recall the first and second letters. While being shown the third letter in the string, subjects pressed the button for the letter seen two letters previously (the first letter seen four seconds prior). While viewing the fourth letter, subjects pressed the button for the next letter seen two letters previously (second letter in the series). 

This continuous task was designed for subjects to engage their working memory on a string, rearrange the letters, and disengage their working memory to re-focus their attention to the next letter. 

Subjects have individual strategies for remembering single letters in series (e.g., A-B-C-D) or through “chunks” (AB-BC-CD, or ABC-BCD). Each one-minute block was repeated five times. Final data from this five-minute task was in the form of time series scans of 45 letters for the first 0-Back stimulus response, and 35 letters for the second letter back (2-Back task) response. This corresponded to five blocks of seven responses each.

### 2.4. Data Collection 

Neuroimaging was performed in a Siemens 3T Tim Trio scanner equipped with a transmit-receive body coil and a commercial 12-channel head coil array. Structural 3D T1-weighted Magnetization Prepared Rapid Acquisition Gradient Echo (MPRAGE) image parameters were TR/TE = 1900/2.52 ms, TI = 900 ms, field-of-view (FoV) = 250 mm, 176 slices, slice resolution = 1.0 mm, and voxel size 1 × 1 × 1 mm. Functional T2*-weighted gradient-echo planar imaging (EPI) parameters were number of slices = 47, TR/TE = 2000/30 ms, flip angle = 90°, matrix size = 64 × 64, FoV = 205 mm^2^, and voxel siz = 3.2 mm^2^ (isotropic). 

The CONN version 17 toolbox was used to pre-process BOLD data according to the default procedure [23]. These pre-processing steps included: (a) time alignment through slice timing correction (STC), (b) smoothing with a stationary Gaussian filter of 6 mm full-width at half maximum (FWHM), (c) spatial pre-processing and positioning of the resulting fMRI images to the Montreal Neurological Institute (MNI) standard stereotactic space, (d) segmentation and outlier detection on Artifact Detection Tools, (e) rectification of the functional images to remove warping [24], Spatial normalization resulted in a voxel size of 2.0 mm^3^ (isotropic). Preprocessed EPI data for each subject was modeled with SPM12 software to the following: instruction, fixation, 0-Back, and 2-Back [25]. We determined that the contrast of interest was 2-Back > 0-Back and opted for a one-sample t-test with motion parameters such as translation and rotation to be covariates of no-interest. Using this 2-Back > 0-Back condition allowed us to identify voxels that had increased significance in activation during the high cognitive 2-Back load over the low cognitive 0-Back load. 

### 2.5. Feature Extraction

A customized MATLAB program that used functions from both SPM12 and xjView 9.6 grouped voxels into AAL regions as mapped out by the MNI coordinates extracted from resultant functional T-statistical maps [26]. One-sample T values were generated from BOLD data based on the relative activation in the 2-Back > 0-Back condition of each individual. We determined the optimal threshold of voxel activation by plotting the total voxels activated as a function of T values as described in a previously [14]. It was determined that voxels with T values exceeding 3.17 (*p* < 0.001 uncorrected) maintained statistical significance while allowing a large number of voxels per person to remain in consideration. The Automated Anatomical Labeling (AAL) atlas (Appendix A) was used as an anatomical map and significant voxels were grouped into corresponding regions [15]. Each AAL region was considered a separate feature for modeling purposes. Total significant voxels for each AAL region were the “features” or independent variables that were fed into the model and analyzed for the purpose of model building. The full AAL depicting regions, centers of mass, and voxels per region are detailed in the Appendix A [27]. 

A multistep feature reduction process was used to determine the number of significant AAL regions, or features. In the first step, multicollinearity was assessed by finding the Pearson correlation coefficients between levels of BOLD activation for all subjects in all AAL regions. Every AAL region was compared against every other AAL region to create a matrix of correlation coefficients. A threshold of R < 0.9 was implemented in order to eliminate or combine highly correlated inputs from the model. If a region had a correlation coefficient of > 0.9 with any other region, it would be assumed that it could be linearly predicted from another coefficient and thus the resulting regions would need to be combined or compared and then removed. Although multicollinearity may not impact the accuracy or predictive power of a model, it can affect the coefficients and calculations of individual predictors, rendering the results for any single variable invalid [28,29]. For instances of perfect multicollinearity where one individual predictor is the perfect combination of the remainder, the design matrix has less than one full rank and therefore cannot be inverted, which can eliminate the ordinary least squares estimator [30]. The presence of multicollinearity in a final model can cause excessively large standard errors for coefficients, inaccurate predictions of the model, and models that overfit the data [31,32]. Tests to ensure there were no correlations between variables were repeated three times on the training set, testing set, and combined training and testing set (full dataset) to make sure no regions of perfect multicollinearity persisted across any of the data. 

The second step of the feature selection process sought to reduce the total number of features using an iterative recursive feature elimination algorithm. Recursive feature elimination (RFE) performs a greedy search algorithm that iteratively cycles through inputs and determines the best and worst performing inputs. After all inputs are explored, it seeks to reduce the model to the fewest possible inputs as the principles of Occam’s Razor dictates that simplest models lead to the most accurate results [33]. Although this greedy search is similar in nature to a stepwise logistic regression, it was selected to optimize p-values, standard errors, confidence intervals, bias in R^2^ values, and create un-biased parameter estimates. It was important to use RFE instead of a stepwise logistic regression because a stepwise algorithm can exacerbate collinearity problems in small datasets, and this was a very small sample size. The scikit-learn python package default recursive feature elimination (RFE) algorithm was used for this step [34]. The weakest features were removed by an iterative process until an optimal number of features and accuracy were obtained. Iteratively eliminating a few inputs every loop reduces the potential for overfitting and decreases the total number of variables with inter-dependencies to produce an improved model.

The third step determined the optimal modeling method and partitioned the data into appropriate training and testing sets. In addition to testing logistic regression, the authors evaluated a series of other techniques including a support vector machine (SVM), neural net, decision tree, and random forest. The logistic regression was optimal because of its simplicity, generalizability, and ability to predict a binary dependent variable from multiple input independent variables [35,36]. The logistic was ultimately selected due to its ability to consistently predict GWI from an SC with similar results using the same variables upon repeated trials in addition to its speed. The Support Vector Machine for example ran for three days on one trial to no conclusion, and the Random Forest was difficult to replicate in practice. For example, if this algorithm was to go into effect in practice in the medical field, it would be difficult to code the potential 30,000 branches of a random forest into a standard fMRI analyzer. The logistic regression provided a consistent, generalizable, and easily implementable alternative. Although the logistic regression is described here, results from alternate modeling techniques are available in the Appendix A. The data were partitioned into a series of training and test sets from 50:50 to 90:10 and iteratively tested to select the proportions that optimized accuracy. The optimal split was a 70:30 randomly stratified training to testing set. We determined this ratio through evaluation of the ultimate predictive power of the model. For example, if a model showed a high accuracy (e.g., 80%) initially, that could not be replicated on repeated attempts, it would be determined that this was likely too generous of a split and caused overfitting and a lack of generalizability. The ratio of 70:30 in the final model gave similar results upon repeated trials. 

Logistic regression model algorithms available from scikit learn and the statsmodels python package were both evaluated in this process. Age, gender, and Body Mass Index (BMI) were included as independent covariates in all regressions to account for differences amongst groups. 

It is important to note that this study did face potential biases due to the small sample size. The 70:30 split provided a substantially larger amount of samples in the training set; however, depending on the ultimate make-up of the testing set could have also been heavily overfit. Further studies with more data would eliminate this problem. 

Logistic regression seeks to estimate coefficients for the logarithm of the odds (log-odds) pertaining to an outcome variable (GWI vs. control status) based on the linear combination of independent variables (voxel counts per AAL region) [37]. It seeks to understand how each corresponding coefficient for an input variable can be “regressed” from the data [38]. Logistic coefficients are obtained by the logistic regression model procedure, which seeks to fit coefficients to input variables according to the logit equation. The general form of the logit equation is
p(x) = 1/(1 + e^−(β0 + β1 × 1 + β2 × 2 …+ βixi)^)
where β_0_ is the intercept, which is a constant term, and β_1_ is the corresponding coefficient for the first variable x_1_, β_2_ is the coefficient for variable x_2_, and β_i_ represents the coefficient assigned to all additional variables [39]. Coefficients are determined by first receiving a coefficient (β) that is next reduced according to the principles of stochastic gradient descent until the highest possible accuracy is obtained from a potential model. During the Stochastic gradient descent reduction process, steps are taken in an iterative fashion that are proportional to the negative gradient of the function at every point and then continued until a minimum is met [40]. The probability (p(x)) calculated for each subject was used to predict GWI versus control status. 

Each set of coefficients from the iterative training sets were applied to their test sets to assess model accuracy and generalizability. Model accuracy was determined by overall false positive rate (accuracy) from the validation data (testing set). Model was further validated using 10-fold cross-validation (k-cross validation with k = 10) and 10 smaller sampled subgroups drawn at random from the 30% validation data (testing set; out-of-sample testing). This process of cross validation reuses these smaller sampled subgroups on the validation data only to validate generalizability by recalculating false positive rates. The final result is averaged across each subgroup to obtain the ultimate accuracy. Due to the small sample size of the data, the total number of cross-validated subgroups was selected to be 10 to provide a generalizable outcome and potentially reduce overfitting. Results from a cross-validation procedure provide a more accurate method of assessing a model’s predictive performance and capacity on potential future data [41]. However, even with the use of cross-validation there is still room for significant bias and over-fitting of the results that more data would alleviate. Multiple studies have detailed how cross-validation can be impacted by over-fitting [42]. The authors would like to note that this bias exists and indicate the need for more data.

The use of cross validation ensured that the predictive model was first trained on multiple different re-sampled ratios of training: testing sets, tested on testing (validation) data separate from the training data that was re-sampled, and was re-built in an iterative fashion until the remaining set of input features and corresponding model coefficients allowed a high degree of both accuracy and generalizability. The final result of the model build was a predictive indicator that showed how a group of independent variables (in this case total voxels confined to each AAL region) was able to predict the dependent binary outcome (GWI vs. control). To correct for multiple comparisons for multiple regions the Sidak method was used and evaluated within the python procedure of the logistic regression [43].

To assess statistical significance of the final predictive model, a “Shuffle Test” was then used for both Days 1 and 2. To run a Shuffle Test, the outcome labels on all subjects corresponding to GWI or SC were re-shuffled in python using a built-in randomization function. This newly labeled data was then passed through the model build process 1000 times to test both if the original accuracy and result could occur a majority of times due to random chance. It should be noted that the randomized sample was also split into a 70% training set and 30% testing set to mimic model build conditions. To ensure this Shuffle Test worked, the process was repeated up to 10,000 times if no Shuffled run could obtain an accuracy greater than or equal to the reported model accuracy for Day 1 and Day 2. The Shuffle Test attempts to determine the p-value and statistical significance of a predictive model. It can be assumed that a model that obtains accuracy equal to or less than the original 5% of the time would be significant at the *p* < 0.05 level. 

The final model coefficients produced for the Logistic Regression signify the rate of change in the “log odds” of the input feature that changes as the outcome variable changes. In this way, the y-intercept term (β_0_) is therefore representative of the log-odds of the outcome or dependent variable when all of the inputs or predictors are set to 0. The model coefficients in a multivariate model also signify the importance of the variable in the predictive model, and not necessarily the importance of the variable fir the disease itself. As such, the model coefficients cannot be interpreted like the slope coefficients for a simpler linear regression. Instead they are representative of the change in the log-odds relative to one another. To clarify, if the first variable in the series has a corresponding coefficient of β = 1, the variable or x in turn would be multiplied by the log-odds of 1 or 10 (10^1^). Similarly, a coefficient of 2 would cause the variable to be multiplied by 100 (10^2^). Due to the inter-dependency between variables and basis of the multivariate model, although it could have a direct relationship, a coefficient with a negative slope may not necessarily indicate a negative correlation with the dependent variable. The presence of the remaining variables and corresponding coefficients in the multivariate model make it difficult to ascertain this relationship without alternate analysis.

This entire process was repeated to test for the ability to determine GWI orthostatic phenotypes (subgroups) START, STOPP, and POTS [13,23]. Postural tachycardia groups were defined by their postural change in heart rate between supine and standing before and after exercise. STOPP had normal changes (ΔHR 12 ± 5 mean ± SD). POTS developed tachycardia >30 beats per minute on standing both before and after exercise. START had normal heart rate changes before exercise, but events of postural tachycardia of >30 beats per minute in the first 24 h after exercise. Predictive models were similarly sampled, trained, and tested after being subjected to RFE on a variable that segmented the GWI population into START, STOPP, and POTS subgroups. 

The corresponding AAL regions that were predicted to be significant in the final selected model were displayed by selection in Wake Forest PICK ATLAS, then imported into MarsBaR 0.44, and finally shown as color-coded axial slices (MRIcron) [44]. 

The final component of the analysis to validate that the patterns of data were different re-used the original Pearson’s correlation coefficient analysis. The correlation coefficients corresponding to each remaining selected predictive AAL region were displayed for each group (GWI Day 1, SC Day 1, GWI Day 2, SC Day 2) and analyzed for total correlations and for the most correlations. This was repeated for the three orthostatic phenotypes. 

## 3. Results

### 3.1. Subjects

All GWI subjects met CMI and Kansas criteria. Age, gender, and BMI were equivalent between groups (Table 1 and Table 2) [1,3]. Symptoms assessed using a 0 to 4 point anchored ordinal scale were significantly higher for GWI than controls by two-tailed unpaired Student’s *t*-tests. 

### 3.2. Feature Extraction

To account for multicollinearity, Pearson’s correlation coefficient (R) was calculated for every subject across each feature (AAL region). All Pearson’s correlation coefficients were <0.9 (Figure 2). Therefore, all regions were used for the recursive feature elimination steps. Although all of the 117 regions were included, many were later removed in the respective RFE and logistic regression steps.

### 3.3. Feature Selection

Recursive feature elimination identified 29 AAL regions for the Day 1 (Table 3 and Table 4, Figure 3) and 28 for the Day 2 (Table 5) logistic regression models to distinguish GWI from control status. Ten AAL regions were present in both the Day 1 and Day 2 models These were the right insula and thalamus (*Insula_R, Thalamus_R*), right frontal regions (*Frontal_Mid_Orb_R, Frontal_Sup_Orb_R, Paracentral_Lobule_R)*, right calcarine (*Calcarine_R*), bilateral superior temporal gyri (*Temporal_Pole_Sup_L, Temporal_Pole_Sup_R*), and left supramarginal (*SupraMarginal_L*) and left cerebellar crus 1 (*Cerebellum_Crus1_L*). These represented a subgroup of regions with differential activation between GWI and control groups that were present in both models before and after exercise.

The Day 1 model included these 10 regions plus 19 other regions that were selected only on Day 1. GWI was distinguished from control status by the combination of these 29 regions before exercise. Of note was activation of the right insula (*Insula_R*) that is implicated in salience and the right thalamus (*Thalamus_R*) that is heavily involved in many processes [15,45]. Left and right frontal regions that may overlap with the anterior default mode network (DMN) included the bilateral superior frontal gyrus, the bilateral superior frontal gyrus medial orbital, the left inferior gyrus triangular part, the right inferior frontal gyrus, and the gyrus rectus (*Frontal_Sup_L, Frontal_Inf_Tri_L, Frontal_Mid_Orb_L, Frontal_Mid_Orb_R, Frontal_Sup_Orb_R, Frontal_Inf_Oper_R and Rectus_R*) [45]. N-Back tasks have been associated with activation of a group of caudal right medial and lateral frontal gyrus regions linked to the right supplementary motor area (*Supp_Motor_Area_R, Precentral_R, Paracentral_Lobule_R*) [45]. 

Parietal regions included voxels in the right angular gyrus and left postcentral and supramarginal gyri (*Postcentral_L, SupraMarginal_L, Angular_R*). Bilateral occipital (*Calcarine_L, Calcarine_R, Cuneus_L, Cuneus_R, Occipital_Mid_R, Lingual_R*) and temporal (*Temporal_Sup_L, Temporal_Pole_Sup_L, Temporal_Pole_Sup_R*) regions may have been related to processing of visual and auditory stimuli. Cerebellar contributions included crus 1, crus 2 and lobule 6 (*Cerebellum_Crus1_L, Cerebellum_7b_L, Cerebellum_8_L, Cerebellum_Crus2_R, Cerebellum_6_R*). The largest magnitude coefficients were *Calcarine_L* (0.7595), *Temporal_Pole_Sup_L* (0.5282), Cerebellum_7b_L (0.5321), and *Calcarine_R* (−0.8050). 

The Day 2 model incorporated 18 different regions with the 10 shared between Day 1 and 2. The Day 2 model was remarkable for bilateral thalamus involvement (*Thalamus_L, Thalamus_R*). The anterior salience network was again suggested by cingulate cortex and right anterior insula activation (*Cingulum_Ant_R, Cingulum_Mid_L, Insula_R*) [45]. Portions of the anterior default mode networks were differentially activated (*Frontal_Sup_Medial_L, Frontal_Sup_Medial_R, Frontal_Mid_Orb_L, Frontal_Mid_Orb_R, Frontal_Sup_Orb_R, Olfactory_R*). The left Rolandic operculum and suparmarginal gyrus (*Rolandic_Oper_L, SupraMarginal_L*) and right somatosensory regions (*Paracentral_Lobule_R, Postcentral_R*) were represented. Bilateral visual (*Cuneus_L, Cuneus_R, Occipital_Sup_R, Calcarine_R*) and auditory (*Temporal_Pole_Sup_L, Temporal_Pole_Sup_R, Temporal_Pole_Mid_R)* regions were again recruited but were different from the pattern in the Day 1 model. The left medial temporal lobe was also activated (*ParaHippocampal_L*). The largest single change was the inclusion of many left, right and midline vermis cerebellar regions (*Cerebellum_Crus1_L, Cerebellum_Crus2_L, Cerebellum_4_5_L, Cerebellum_Crus1_R, Cerebellum_7b_R, Cerebellum_8_R, Vermis_3, Vermis_9*). The largest magnitude coefficients were the parahippocampal gyrus (*ParaHippocampal_L*) (0.9032), cerebellum (*Cerebellum_4_5_L*) (0.7197), and vermis (*Vermis_9*) (0.6321) and (*Vermis_3*) (−0.7653). The Day 2 model was notable for the right anterior salience, bilateral thalamus, anterior default, visual, auditory, and cerebellar regions.

The Day 2 model incorporated 18 different regions with the 10 shared between Days 1 and 2. The Day 2 model was remarkable for bilateral thalamus involvement (*Thalamus_L, Thalamus_R*). The anterior salience network was again suggested by cingulate cortex and right anterior insula activation (*Cingulum_Ant_R, Cingulum_Mid_L, Insula_R*) [45]. Portions of the anterior default mode networks were differentially activated (*Frontal_Sup_Medial_L, Frontal_Sup_Medial_R, Frontal_Mid_Orb_L, Frontal_Mid_Orb_R, Frontal_Sup_Orb_R, Olfactory_R*). The left rolandic operculum and suparmarginal gyrus (*Rolandic_Oper_L*, *SupraMarginal_L*) and right somatosensory regions (*Paracentral_Lobule_R*, *Postcentral_R*) were represented. Bilateral visual (*Cuneus_L*, *Cuneus_R*, *Occipital_Sup_R*, *Calcarine_R*) and auditory (*Temporal_Pole_Sup_L*, *Temporal_Pole_Sup_R*, *Temporal_Pole_Mid_R*) regions were again recruited but were different from the pattern in the Day 1 model. The left medial temporal lobe was also activated (*ParaHippocampal_L*). The largest single change was the inclusion of many left, right and midline vermis cerebellar regions (*Cerebellum_Crus1_L, Cerebellum_Crus2_L, Cerebellum_4_5_L, Cerebellum_Crus1_R, Cerebellum_7b_R, Cerebellum_8_R, Vermis_3, Vermis_9*). The largest magnitude coefficients were the left parahippocampal gyrus (*ParaHippocampal_L)* (0.9032), cerebellum (*Cerebellum_4_5_L)* (0.7197), and vermis (*Vermis_9)* (0.6321) and (*Vermis_3)* (−0.7653). The Day 2 model was notable for the right anterior salience, bilateral thalamus, anterior default, visual, auditory, and cerebellar regions.

### 3.4. Model Results and Validation

The features were fed into the logistic regression model to determine accuracy and predictive power for GWI vs. SC. The Day 1 model had similar accuracies for training (70.6%) and 10-fold cross-validation (70.5%) (Table 4). On Day 2, the accuracy increased to 85.3% indicating that exercise accentuated the multivariate pattern of AAL regions that distinguished GWI and control groups. Accuracy was lower for the smaller cross-validation set (63.4%) but did not drop to random accuracy (50%), which suggested that the Day 2 training model did not overfit its data. 

Table 6 shows the sensitivity and specificity for the cross validated testing sets.

Performance of a Shuffle Test where labels were scrambled, and the model build was repeated and reproduced the target accuracy of 70% or greater on 49 of the 1000 shuffled runs for the first day. As both results for the Day 1 model and cross validated result were shown to have an accuracy greater than 70%, it can be assumed that the pre-exercise model was significant at the *p* < 0.05 level. Performance of the Shuffle Test similarly on the Day 2 results produced accuracies between 60% and 85% on 40 of 1000 and 0 of 1000 shuffled runs, respectively. This indicated that tis predictive model for Day 2 is statistically significant at the *p* < 0.05 level as well.

It is important to recall that it is the multivariate pattern of regions that predicts disease. Individual regions acting alone are not easily discernible from the logistic regression model, but it was of note that bilateral medial frontal superior gyri, left middle, and right anterior cingulate gyri, and numerous cerebellar regions contributed to the difference between GWI and controls on Day 2 after the submaximal exercise provocations. This statistical model suggested that exercise caused changes in brain activation during the 2-Back cognitive task that increased the ability of the logistic regression model to differentiate GWI from control subjects. 

Logistic regression model based on 49 AAL regions distinguished the START, STOPP and POTS subsets from each other on Day 1 (Table 7). Only 29 regions were selected by recursive elimination for the Day 2 model. The numbers of voxels per region varied between subsets and Days 1 and 2 but were not significantly different between groups by ANOVA. Instead, it was the combinations of regions that determined the differences in the models. It was notable that the Day 2 model was composed of ROIs that were a subset of the Day 1 model; no new ROIs were incorporated. The combination of regions on Day 2 may represent a core territory for distinguishing the three GWI postural tachycardia groups.

Overall, the logistic regression and cross-validation models were resilient and maintained their accuracies at 0.625 to 0.690 (Table 8).

The resultant Shuffle Test achieved accuracies of 62% on 45 and 43 of the respective 1000 shuffled runs for Days 1 and 2. The average accuracy was only 35% on each shuffled run, suggesting the accuracies were significant at the *p* < 0.05 level and unlikely to be obtained by random chance alone. 

The groupings and corresponding patterns of Pearson’s Correlation Coefficients were compared for all 117 AAL regions (Figure 4). Qualitative analysis indicated that these patterns differed between both groups and days. The highest number of positive correlations between anatomical AAL regions both for correlations overall and for R ≥ 0.7 occurred in the sedentary control group on the first day. These results potentially indicated that the control subjects were focused on the task on Day 1. Similar to GWI, the control group showed fewer correlations after exercise on Day 2, which indicated they had potentially learned the task and were presenting with automaticity and a lower level of focus required to complete the working memory task. 

The plot showing the GWI group showed fewer correlations overall and ≥0.7 on the fMRI scan performed before exercise. They also exhibited different patterns from the sedentary control group on both days. GWI had limited similarity in correlations and corresponding activations between the two days. Visual qualitative analysis of this data and quantitative analysis by pursuing correlations ≥0.7 between the AAL regions further supports the results of the logistic regression analysis. This analysis also indicates there are differences in connectivity between GWI before and after exercise and the sedentary control. From these results it can be postulated that further analysis of functional connectivity in fMRI data between GWI and a SC both before and after exercise will show differences. This can potentially be explained by the post−exertional malaise [46].

The GWI group was subdivided into START, STOPP, and POTS groups, and the Pearson correlation matrices were compared (Figure 5). POTS had the most correlations with R > 0.7 on both days. Patterns for these subgroups were also dissimilar from SC on Days 1 and 2. These data suggest that the three GWI orthostatic tachycardia phenotypes and control subjects had different networks of activated brain regions and lend credibility to the hypothesis that these phenotypes have distinct neurological patterns that contribute to their pathologies of GWI.

## 4. Discussion

The strategy of recursive feature elimination and logistic regression generated models that distinguished Gulf War Illness (GWI) from sedentary control (SC) subjects on Day 1 (pre−submaximal exercise) and Day 2 (post−submaximal exercise).

The three orthostatic tachycardia subgroups of GWI subjects were also distinguished on Day 1 and Day 2. However, the model was more accurate on Day 2 and incorporated fewer AAL regions.

The Day 1 (pre−submaximal exercise) results indicate that different brain regions were activated and had hemodynamic (BOLD) responses during the N−back memory paradigm in veterans with GWI compared to SC subjects. On Day 2, the N−Back testing activated different patterns of regions indicating that the exercise stress tests caused changes in the patterns of brain activation but were still able to differentiate GWI from SC. The differences between Days 1 and 2 infer that exercise altered the balance of relative activation between GWI and SC but do allow for conclusions about the effects of exercise on specific brain regions or functions in control subjects.

This study was limited in that it only obtained data for 111 subjects. The number of activated voxels per AAL region was relatively small and attributable to the small sample size. Larger studies would produce more accurate predictive models for classification of GWI from sedentary control subjects. Recursive feature elimination and logistic regression models are inherently limited in that they can seek to maximize the number of individuals selected, or minimize the errors surrounding the coefficients on model inputs. For this study, we used the default sci−kit learn threshold, but by altering this parameter, one could potentially produce a less accurate model that encompassed more subjects or a more accurate model that encompassed fewer subjects. The logistic regression does not directly take into account the magnitude of activation of individual voxels in the 2−Back > 0−Back condition and instead infers it based on total overall activated voxels. Input variables were binned at a threshold (*t* > 3.17 corresponding to *p* < 0.001 uncorrected) and due to Gaussian smoothing in the SPM preprocessing, regions with larger numbers of activated voxels had higher overall magnitudes of activation.

This model is further limited in that it can identify an entire pattern of difference between groups but does not indicate that any single region is significant enough to distinguish between subjects. However, the complex of differentially activated regions may contain smaller discrete cortical regions that had significant changes in the magnitude of their local BOLD signals. Region of interest analysis or other methods may be useful to discover these potential biomarker regions. The multivariate pattern identified by this analysis signifies that this type of methodology and use of machine learning could be useful as a diagnostic indicator of GWI from a sedentary control. Future studies could use this pattern to distinguish Gulf War illness or this methodology to differentiate other disorders. 

## 5. Conclusions

Logistic regression analysis identified patterns of regional brain activation that distinguished Gulf War Illness (GWI) and sedentary control (SC) with accuracies of 70.6% on Day 1 and 85.3% on Day 2. Logistic Regression was also able to differentiate the three orthostatic phenotypes of START, STOPP, and POTS at 62% accuracy on both days. The results of this study imply that classification of a multivariate pattern of activated regions can potentially be used as a clinical biomarker for GWI diagnosis. 

## Figures and Tables

**Figure 1 brainsci-10-00319-f001:**
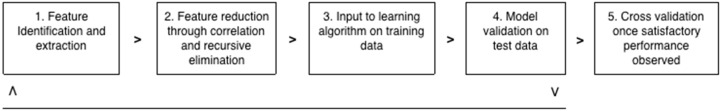
Framework of our predictive model build.

**Figure 2 brainsci-10-00319-f002:**
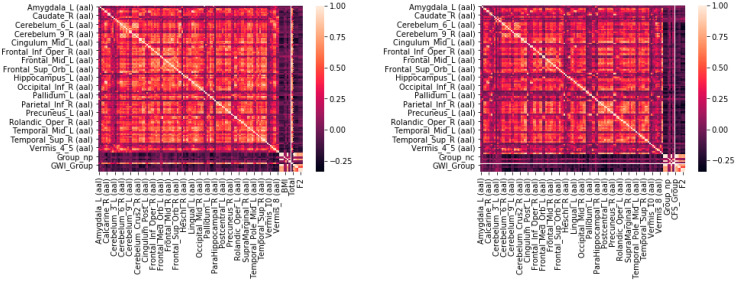
Heatmap of Pearson’s Correlation Coefficients. The numbers of significantly activated voxels (*t* > 3.17, *p* < 0.0001) in each Automated Anatomical Labeling (AAL) region for all subjects were compared on Day 1 (pre-submaximal exercise, left) and Day 2 (post-submaximal exercise, right).

**Figure 3 brainsci-10-00319-f003:**
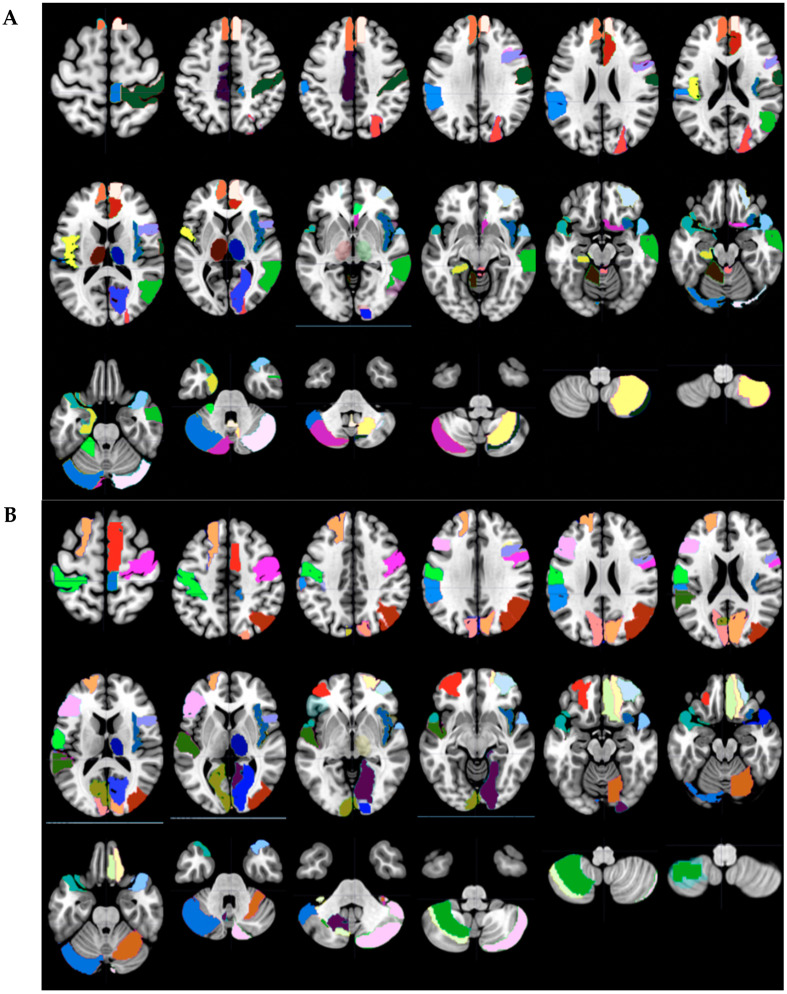
Automated Anatomical Labeling (AAL) regions that distinguished GWI from control subjects on the Day 1 and 2 logistic regression models. Axial slices (left side on left of each slice) were shown for (**A**) Day 1: Pre-Submaximal Exercise Comparison of N-Back > 0-Back test between GWI and Sedentary Control and (**B**) Day 2: Post Submaximal Exercise Comparison of N-Back > 0-Back test between GWI and Sedentary Control. Ten AAL regions that were in both the Day 1 and Day 2 logistic models were colored dark blue (see SOM Appendix A and Appendix A for detailed descriptions of each region) and included the right frontal pole, left and right temporal poles, right insula, right thalamus and cerebellar lobules. The other colors have no specific significance. (**A**) Nineteen additional regions differentiated GWI from control on Day 1, and included right superior frontal, bilateral orbital frontal, left dorsolateral prefrontal, left parietal, and visual regions. (**B**) Eighteen additional regions were identified on Day 2 and included medial frontal gyri in the frontal pole, left middle cingulate, left thalamus, right occipital and temporal regions, and cerebellar regions.

**Figure 4 brainsci-10-00319-f004:**
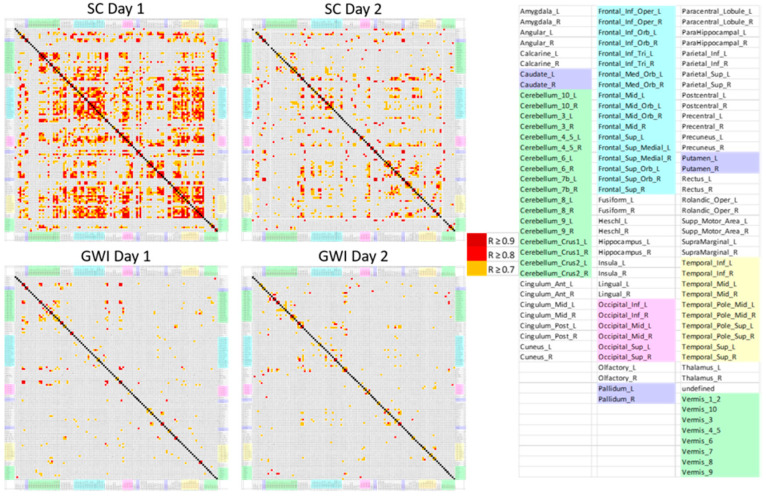
Correlation matrices for the numbers of activated voxels per AAL region. AAL regions were arranged in alphabetical order on the x− and y−axes. The number of significant Pearson correlations (R > 0.7 in orange, R > 0.8 in red, R > 0.9 in dark red) was higher for the SC group on Day 1 and decreased on Day 2. However, GWI had few correlations between regions on either Day 1 or 2. There were no negative correlations (R < −0.5).

**Figure 5 brainsci-10-00319-f005:**
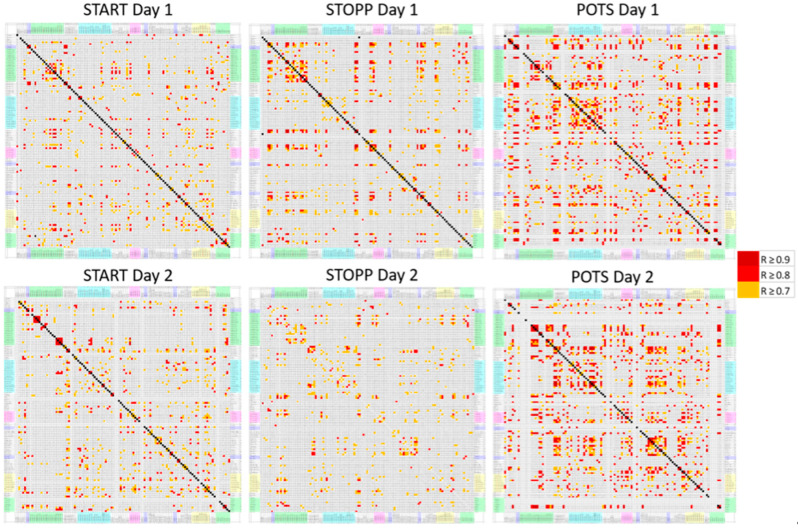
Correlation matrices for the numbers of activated voxels per AAL region. AAL regions were arranged in alphabetical order on the x− and y−axes. The number of significant Pearson correlations (R > 0.7 in orange, R > 0.8 in red, R > 0.9 in dark red) was relatively higher for the POTS group on Days 1 and 2 compared to the other groups. There were no negative correlations (R < −0.5).

**Table 1 brainsci-10-00319-t001:** Demographics (mean ± SD).

Group	SC	GWI
N	31	80
Age	43.9 ± 16.3	46.9 ± 7.8
BMI	28.4 ± 4.5	29.6 ± 5.6
Male	19 (61.3%)	59 (73.8%)
White	23 (74.2%)	64 (80.0%)
CFS Symptom Severity Scores *^,†^		
Fatigue	1.15 ± 1.0	3.5 ± 0.7
Memory and concentration	1.0 ± 1.2	3.1 ± 0.8
Sore throat	0.2 ± 0.6	1.4 ± 1.2
Sore lymph nodes	0.1 ± 0.4	1.5 ± 1.3
Muscle pain	0.6 ± 0.9	3.1 ± 1.0
Joint pain	0.8 ± 1.0	3.2 ± 1.0
Headaches	1.0 ± 1.3	2.7 ± 1.2
Sleep	1.7 ± 1. 4	3.5 ± 0.8
Exertional exhaustion	0.5 ± 1.0	3.3 ± 1.0

* Scale: 0 = none, 1 = trivial, 2 = mild, 3 = moderate, 4 = severe. ^†^ FDR < 0.05 for each item to correct for multiple comparisons.

**Table 2 brainsci-10-00319-t002:** The Gulf War Illness (GWI) population exhibited similar age, body mass index (BMI), and gender distributions.

Group	N (Male)	Age (Mean ± SD)	BMI (Mean ± SD)
START	23 (19)	45.4 ± 8.0	28.0 ± 4.3
STOPP	46 (31)	48.1 ± 7.2	30.2 ± 5.9
POTS	11 (9)	45.7 ± 6.0	30.4 ± 6.1

**Table 3 brainsci-10-00319-t003:** Combinations of thalamus and cerebellum AAL regions predicting GWI versus control status. Logistic regression model coefficients and voxels per AAL region were shown for models that predicted GWI versus control status prior to (Day 1) and after (Day 2) exercise.

**Cerebellum**			
		Day 1	Day 2
ID	AAL Region	Coefficient	Coefficient
	Intercept	0.36	1.87
**Thalamus**			
77	*Thalamus_L*		−0.2715
78	*Thalamus_R*	−0.1715	0.0304
**Left Cerebellum Lobules**			
91	*Cerebellum_Crus1_L*	−0.0297	−0.0707
101	*Cerebellum_7b_L*	0.5321	
103	*Cerebellum_8_L*	−0.4121	
93	*Cerebellum_Crus2_L*		0.0672
97	*Cerebellum_4_5_L*		0.7197
**Right Cerebellum Lobules**			
94	*Cerebellum_Crus2_R*	−0.2809	
100	*Cerebellum_6_R*	0.0264	
92	*Cerebellum_Crus1_R*		0.0134
102	*Cerebellum_7b_R*		0.1074
104	*Cerebellum_8_R*		−0.0562
**Midline**			
110	*Vermis_3*		−0.7653
115	*Vermis_9*		0.6321

**Table 4 brainsci-10-00319-t004:** Combinations of cerebral AAL regions predicting GWI versus control status. Logistic regression model coefficients and voxels per AAL region were shown for models that predicted GWI versus control status prior to (Day 1) and after (Day 2) exercise. *p* < 0.05 for all reported variables once corrected using Sidak correction.

		Day 1	Day 2
ID	AAL Region	Coefficient	Coefficient
	Intercept	0.36	1.87
	Left Hemisphere		
63	*SupraMarginal_L*	0.1442	0.0213
83	*Temporal_Pole_Sup_L*	0.038	−0.0213
9	*Frontal_Mid_Orb_L*	0.2989	
3	*Frontal_Sup_L*	−0.0646	
13	*Frontal_Inf_Tri_L*	0.0494	
57	*Postcentral_L*	−0.1314	
81	*Temporal_Sup_L*	0.5282	
45	*Cuneus_L*	0.046	
43	*Calcarine_L*	0.7595	
23	*Frontal_Sup_Medial_L*		−0.0385
33	*Cingulum_Mid_L*		0.0945
17	*Rolandic_Oper_L*		0.0205
39	*ParaHippocampal_L*		0.9032
**Right Hemisphere**			
30	*Insula_R*	−0.0368	−0.0131
10	*Frontal_Mid_Orb_R*	−0.0168	0.0382
6	*Frontal_Sup_Orb_R*	0.0627	0.0245
70	*Paracentral_Lobule_R*	−0.4414	−0.2976
44	*Calcarine_R*	−0.805	−0.0535
84	*Temporal_Pole_Sup_R*	−0.3209	0.2698
28	*Rectus_R*	−0.366	
12	*Frontal_Inf_Oper_R*	−0.052	
20	*Supp_Motor_Area_R*	−0.0422	
2	*Precentral_R*	0.1018	
66	*Angular_R*	0.0399	
46	*Cuneus_R*	−0.1037	
52	*Occipital_Mid_R*	−0.0461	
48	*Lingual_R*	−0.3924	
32	*Cingulum_Ant_R*		−0.0409
22	*Olfactory_R*		−0.4417
24	*Frontal_Sup_Medial_R*		0.0653
58	*Postcentral_R*		−0.0247
50	*Occipital_Sup_R*		−0.0219
88	*Temporal_Pole_Mid_R*		−0.3901

**Table 5 brainsci-10-00319-t005:** Model accuracy on Days 1 (pre-submaximal exercise) and 2 (post submaximal exercise) and cross validation results. These results indicate that the Day 2 model, or model created after exercise, had a higher degree of accuracy that the Day 1 model. Although both were able to differentiate GWI from an SC, the Day 2 model performed slightly better.

	Day 1 (Pre-Submaximal Exercise)	Day 2 (Post-Submaximal Exercise)
Accuracy of logistic regression classifier on test set:	70.60%	85.30%
10-fold cross validation average accuracy:	70.50%	63.40%

**Table 6 brainsci-10-00319-t006:** Matrix of total reported subject results and general reported statistics for actual totals of subjects captured by model prediction on test set. Sensitivity and specificity are reported for the most accurate performing non-cross validated model (70.6% on Day 1 and 85.3% on Day 2).

	Sensitivity	Specificity	PPV	NPV
	**(TP/(TP + FN))**	**(TN/(TN + FP))**	**(TP/(TP + FP))**	**(TN/(TN + FN))**
Day 1 (Pre-submaximal Exercise)	46%	90%	77%	69%
Day 2 (Post-Submaximal Exercise)	71%	33%	71%	60%

**Table 7 brainsci-10-00319-t007:** AAL regions in logistic regression models that distinguished GWI START, STOPP, and POTS subsets on Day 1 and Day 2. Recursive feature elimination identified regions that significantly separated the GWI subsets on Day 1 and Day 2. The models identified the combinations of regions with different numbers of significant voxels (*t* > 3.17, *p* < 0.001 for the 2 > 0-back condition) per ROI. Day 2 regions were a subset of Day 1.

L R	AAL Index	AAL Region	Day 1 Coefficients			Day 2 Coefficients		
			START	STOPP	POTS	START	STOPP	POTS
		Intercept	−0.3254	−0.4127	0.3718	−0.1666	−0.0894	0.0274
L	5	*Frontal Sup Orb L*	0.0211	−0.0407	0.0181			
L	9	*Frontal Mid Orb L*	0.0164	−0.1065	0.0126	−0.4932	0.0184	0.0305
L	11	*Frontal Inf Oper L*	−0.105	−0.1363	0.2162			
L	7	*Frontal Mid L*	−0.0231	0.0383	−0.0026	−0.0147	0.0587	−0.0739
L	3	*Frontal Sup L*	−0.0512	−0.1179	0.1449	0.1265	−0.1338	0.1886
L	19	*Supp Motor Area L*	0.1117	−0.0605	0.0136	−0.1579	−0.0389	0.0982
L	69	*Paracentral Lobule L*	−0.1134	0.1053	−0.0204	−0.2408	−0.3127	0.3879
L	1	*Precentral L*	0.0928	−0.0498	−0.0864			
L	57	*Postcentral L*	0.0961	0.0871	−0.2163	0.3113	0.2695	−0.2883
L	63	*SupraMarginal L*	0.0877	−0.1433	−0.1304	−0.071	0.1057	−0.0139
L	65	*Angular L*	−0.0895	0.0158	0.1133	0.2232	0.0589	−0.0733
L	51	*Occipital Mid L*	−0.0548	0.1066	−0.0938	0.0097	0.0655	−0.0653
L	43	*Calcarine L*	−0.0111	−0.0567	0.0071			
L	83	*Temporal Pole Sup L*	−0.1597	0.1995	−0.1059			
L	81	*Temporal Sup L*	−0.088	−0.0438	0.2625	−0.0577	−0.2622	0.1185
L	89	*Temporal Inf L*	−0.076	−0.0125	−0.0124	0.0575	0.0984	−0.1835
L	31	*Cingulum Ant L*	0.0566	0.0959	−0.0963			
L	33	*Cingulum Mid L*	0.0297	0.0179	0.0812	0.1534	−0.2875	0.2176
L	71	*Caudate L*	−0.0058	0.0442	−0.0503			
L	75	*Pallidum L*	−0.0224	−0.0114	0.0408			
L	93	*Cerebellum Crus2 L*	−0.0164	0.1204	−0.1847	−0.3072	−0.0044	0.2066
L	99	*Cerebellum 6 L*	0.0299	0.0562	−0.1019	0.2718	−0.1082	−0.0449
L	103	*Cerebellum 8 L*	−0.0598	0.0566	−0.1381	−0.2062	0.0697	−0.1023
R	30	*Insula R*	−0.0488	0.0077	−0.1392	0.0386	0.1679	−0.1451
R	26	*Frontal Med Orb R*	−0.0304	0.0593	−0.0612			
R	16	*Frontal Inf Orb R*	−0.065	0.0401	0.021	−0.1562	0.2341	−0.1199
R	10	*Frontal Mid Orb R*	0.0037	−0.0985	0.1253	0.1542	−0.0252	−0.1217
R	6	*Frontal Sup Orb R*	0.0092	0.0779	0.0805			
R	14	*Frontal Inf Tri R*	0.0015	−0.0007	0.0023	−0.2509	−0.2165	0.2417
R	4	*Frontal Sup R*	0.0175	−0.0247	−0.0211	0.0231	−0.2045	0.126
R	24	*Frontal Sup Medial R*	0.0033	0.0112	−0.1266	−0.4661	0.0834	0.0898
R	20	*Supp Motor Area R*	−0.1253	0.0302	0.1129	0.0463	0.0231	−0.1466
R	2	*Precentral R*	−0.0915	0.058	0.0434	0.0509	0.113	−0.0868
R	58	*Postcentral R*	0.0028	−0.1191	0.0851			
R	18	*Rolandic Oper R*	0.0442	−0.0095	−0.016	0.0634	−0.1134	−0.0481
R	54	*Occipital Inf R*	0.005	−0.0153	−0.0269			
R	48	*Lingual R*	0.0194	−0.0209	0.0209	0.2479	0.0873	−0.0667
R	56	*Fusiform R*	−0.0112	−0.0652	0.1056			
R	82	*Temporal Sup R*	−0.0268	0.0053	−0.062	−0.0754	0.3209	−0.1277
R	34	*Cingulum Mid R*	0.1826	0.0308	−0.166	0.2548	0.0095	−0.088
R	36	*Cingulum Post R*	−0.0018	0.0007	0.019			
R	72	*Caudate R*	−0.0057	0.0647	0.0026			
R	100	*Cerebellum 6 R*	−0.1639	0.0832	0.0797			
R	102	*Cerebellum 7b R*	−0.0342	−0.051	0.1401	−0.2785	−0.0127	0.0302
R	104	*Cerebellum 8 R*	−0.072	−0.0188	0.089	−0.1158	−0.118	0.1482
R	106	*Cerebellum 9 R*	−0.0155	0.0817	−0.1665			
z	111	Vermis 4 5	−0.0042	−0.0033	−0.0763			

**Table 8 brainsci-10-00319-t008:** Model accuracy on Days 1 and 2, cross validation results, and sensitivity and specificity information.

Day 1	Day 2
Accuracy of logistic regression classifier on test set: 62.5%	Accuracy of logistic regression classifier on test set: 62.5%
10−fold cross validation average accuracy: 66.7%	10−fold cross validation average accuracy: 69%

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
