# Peer review of "Logistic Regression Algorithm Differentiates Gulf War Illness (GWI) Functional Magnetic Resonance Imaging (fMRI) Data from a Sedentary Control"

_brainsci, 2020, doi:10.3390/brainsci10050319_

Round 1

Reviewer 1 Report

Machine learning in neuroimaging is emerging as a promising tool for providing diagnostic biomarkers. Although imaging data from individual are affected by diseases in various ways, it is often difficult to find definitive measures with sufficient sensitivity and specificity for diagnosing and classifying an individual. Machine learning is helpful to improve high classification performance of individuals on the basis of multivariate pattern analysis. In the present study, logistic regression estimation was applied to the analysis of fMRI data collected from 80 patients with Gulf War Illness (GWI) and 31 control subjects to determine how a subset of regional activations in the 2-back working memory task could jointly separate the GWI patients from controls with high accuracy. The fMRI data were obtained in two separate sessions on different days with Day 1's data collected before physical exercise and Day 2's data collected after exercise. The proposed logistic regression model was able to differentiate 3 subtypes of GWI from the controls with good accuracy. Exercise served to improve the classification accuracy.

Overall, the manuscript is well written, and the importance of the machine learning approach is properly explained. I have some technical concerns over the experimental design, model validation approach, and data interpretation.

  1. The inclusion and exclusion criteria for the two groups of participants can be more specific in addition to what has been provided in the manuscript. While age, gender, BMI (body mass index; please spell out the BMI term when it is used the first time in the main text), and other parameters listed in Table 1 are important factors, there needs to be some basic screening of sensory, cognitive and motor skills. For example, do the subjects have hearing or vision problems? The performance of the working memory task would presumably be affected by these basic skills.
  2. In the method section, it was specified that "the third step determined the optimal modeling method by assessing logistic regression, support vector machine (SVM), neural net, decision tree, and random forest protocols." However, these modelling methods were not properly described in the Introduction section. For instance, how successful were these methods in previous studies of classification using fMRI data? What are the strengths and limitations?  I am not convinced that logistic regression provides the optimal results without actual outcomes from different methods. The authors could have just as well picked other unsupervised or supervised machine learning approaches. 
  3. Acknowledge of the sample size problem needs to be addressed. Notably, the k-fold cross-validation protocol has its own problems although it is one of the most common resampling methods for selecting models to test how machine learning from trained data generalizes to new test data with a small sample size. Since each training set is only (K −1)/K as big as the original training set, the validation set error tends to overestimate the test error for the model on the entire data set. There are at least two problems here. The first is the independence of the training data and test data. Given the small sample size, the more repetitions are performed, the higher the chance of model overfitting you get with sometimes highly accurate but highly biased results. The second is known as the Freedman's Paradox, which shows that in high-dimensional data, some variables can be randomly associated with an outcome variable by chance alone, which can appear to be strongly significant and thus bias the model outcome. In fact, multiple peer-reviewed publications (e.g., Benkeser et al., 2019; Vabalas et al., 2019) have pointed out the need to exercise caution on the use of the k-fold cross-validation approach. For instance, Vabalas demonstrated that "K-fold Cross-Validation (CV) produces strongly biased performance estimates with small sample sizes, and the bias is still evident with sample size of 1000."
    Vabalas, A., Gowen, E., Poliakoff, E., & Casson, A. J. (2019). Machine learning algorithm validation with a limited sample size. PloS one, 14(11), e0224365. https://doi.org/10.1371/journal.pone.0224365
  4. It seems unclear what specificity and sensitivity measures were obtained for the classification results.
  5. Given the classification accuracy results, I think caution must be necessary for claiming that "The multivariate pattern identified by this analysis can stand on its own as a diagnostic indicator of GWI from a sedentary control."
  6. Some correction for multiple comparison needs to be included, and exactly how exercise may help improve the classification results need further explanation.

Author Response

Reviewer One:

Overall, the manuscript is well written, and the importance of the machine learning approach is properly explained. I have some technical concerns over the experimental design, model validation approach, and data interpretation.

Thank you for your comments, we feel that this type of study is important as well. The future of medical imaging and machine learning certainly is intertwined and we appreciate you recognizing the importance of differentiating diseases from neurological data such as GWI that lack physiological markers.

  1. The inclusion and exclusion criteria for the two groups of participants can be more specific in addition to what has been provided in the manuscript. While age, gender, BMI (body mass index; please spell out the BMI term when it is used the first time in the main text), and other parameters listed in Table 1 are important factors, there needs to be some basic screening of sensory, cognitive and motor skills. For example, do the subjects have hearing or vision problems? The performance of the working memory task would presumably be affected by these basic skills.

The authors appreciate the reviewer’s insight into the need for additional screening, demographic, and performance data. BMI is now spelled out upon first instance. Subject data was detailed more extensively in prior publications and we have updated the methods section as such to reflect this. We required moderate or severe levels of symptoms and did not allow mild or trivial symptoms (Fappiano 2020). In regards to screening for hearing or vision skills and the performance on the N-back working task (SC performed better than GWI) a prior study published the results of the test, however this could be included as a supplement in this paper as well if the reviewer so desires. The methods section was updated with the following:

Further information regarding the study protocol, screening, demographics, subject symptoms, subject pain perception, orthostatic measurements, interoceptive complaints, chemical sensitivity questionnaires, and subject quality of life domain data are reported in previous published articles from our group and online as supplementary materials.    It should be noted that all subjects were screened for ability to perform the task prior to fMRI data collection, and were able to practice the N-back memory task until they felt comfortable prior to recording. Performance data from the GWI group was reported in a previous study to be markedly lower (on average about 26% lower) than the sedentary control both before and after exercise.

Rayhan RU, Stevens BW, Raksit MP, et al. Exercise challenge in Gulf War Illness reveals two subgroups with altered brain structure and function. PLoS One. 2013;8(6):e63903. Published 2013 Jun 14. doi:10.1371/journal.pone.0063903

  1. In the method section, it was specified that "the third step determined the optimal modeling method by assessing logistic regression, support vector machine (SVM), neural net, decision tree, and random forest protocols." However, these modelling methods were not properly described in the Introduction section. For instance, how successful were these methods in previous studies of classification using fMRI data? What are the strengths and limitations?  I am not convinced that logistic regression provides the optimal results without actual outcomes from different methods. The authors could have just as well picked other unsupervised or supervised machine learning approaches. 

The authors appreciate the reviewer’s call to attention of this lack in explanation and have updated the text accordingly. It’s true that there are a lot of other modeling techniques out there and a series of them were reviewed before settling on the logistic regression. The methods section has been updated accordingly to reflect this and additional tables will be added to the supplementary material:

The logistic was ultimately selected due to its ability to consistently predict GWI from an SC with similar results using the same variables upon repeated trials in addition to its speed. The Support Vector Machine for example ran for three days on one trial to no conclusion, and the Random Forest was difficult to replicate in practice. For example if this algorithm were to go into effect in practice in the medical field it would be difficult to code the potential 30,000 branches of a random forest into a standard fMRI analyzer. The logistic regression provided a consistent, generalizable, and easily implementable alternative. Although the logistic regression is described here, results from alternate modeling techniques are available in the Supplementary Materials. (Table S2)

  1. Acknowledge of the sample size problem needs to be addressed. Notably, the k-fold cross-validation protocol has its own problems although it is one of the most common resampling methods for selecting models to test how machine learning from trained data generalizes to new test data with a small sample size. Since each training set is only (K −1)/K as big as the original training set, the validation set error tends to overestimate the test error for the model on the entire data set. There are at least two problems here. The first is the independence of the training data and test data. Given the small sample size, the more repetitions are performed, the higher the chance of model overfitting you get with sometimes highly accurate but highly biased results. The second is known as the Freedman's Paradox, which shows that in high-dimensional data, some variables can be randomly associated with an outcome variable by chance alone, which can appear to be strongly significant and thus bias the model outcome. In fact, multiple peer-reviewed publications (e.g., Benkeser et al., 2019; Vabalas et al., 2019) have pointed out the need to exercise caution on the use of the k-fold cross-validation approach. For instance, Vabalas demonstrated that "K-fold Cross-Validation (CV) produces strongly biased performance estimates with small sample sizes, and the bias is still evident with sample size of 1000."

The authors recognize this concern and will note this. The citation was appreciated and has been incorporated into the references. The authors updated the methods section with the following text:

It is important to note that this study did face a lot of potential biases due to the small sample size. The 70:30 split provided a substantially larger amount of samples in the training set, however depending on the ultimate make-up of the testing set, could have also been heavily overfit. Further studies with more data would eliminate this problem.

As listed above, even with the use of cross-validation there is still room for significant bias and over-fitting of the results that more data would alleviate. Multiple studies have detailed how cross-validation can be impacted by over-fitting. The authors would like to note that this bias exists and indicate the need for more data.

  1. It seems unclear what specificity and sensitivity measures were obtained for the classification results.

The authors appreciate this comment as to the lack of explanation surrounding the sensitivity and specificity and have updated the table description to reflect this:

Table 5: Matrix of total reported subject results and general reported statistics for actual totals of subjects captured by model prediction on test set. Sensitivity and Specificity are reported for the most accurate performing non-cross validated model (70.6% on Day 1 and 85.3% on Day 2)

  1. Given the classification accuracy results, I think caution must be necessary for claiming that "The multivariate pattern identified by this analysis can stand on its own as a diagnostic indicator of GWI from a sedentary control."

The authors appreciate this and have updated the line accordingly:

The multivariate pattern identified by this analysis signifies that this type of methodology and use of machine learning could be useful as a diagnostic indicators of GWI from a sedentary control.

  1. Some correction for multiple comparison needs to be included, and exactly how exercise may help improve the classification results need further explanation.

The authors agree correction for multiple comparisons is important and was considered when initially building the model. The authors also appreciate the reviewer calling to attention that this was not included in the initial manuscript. There is a lot of controversy in the literature about whether this needs to be included or if the accuracy alone is sufficient (https://www.ncbi.nlm.nih.gov/pubmed/2081237). However for the purposes of this study the group set the python procedure to only consider coefficients statistically significant to the p< 0.05 level and opted to use the Sidak method to correct for multiple correlations as the Bonferroni method has been listed to be over-conservative in other studies. The following line has been added to the methods section to address this and the Table containing the coefficients has been updated accordingly.

To correct for multiple comparisons for multiple regions the Sidak method was used and evaluated within the python procedure of the logistic regression.

Table 2: p < 0.05 for all reported variables once corrected using Sidak correction.

More about Sidak:

Zbyněk Šidák (1967) Rectangular Confidence Regions for the Means of Multivariate Normal Distributions, Journal of the American Statistical Association, 62:318, 626-633, DOI: 10.1080/01621459.1967.10482935

Blakesley RE, Mazumdar S, Dew MA, et al. Comparisons of methods for multiple hypothesis testing in neuropsychological research. Neuropsychology. 2009;23(2):255‐264. doi:10.1037/a0012850

To correct for multiple comparisons for multiple regions the Sidak method was used and evaluated within the python procedure of the logistic regression.

Table 2: p < 0.05 for all reported variables once corrected using Sidak correction.

The authors appreciate the reviewer’s indication of a need to explain why we chose to use an exercise based task during a test of working memory. Our group previously found that the GWI subgroups were differentiable using fMRI data after the bicycle stress test. As such we had hypothesized the machine learning model would uncover a similar change. What we found instead was that using machine learning was able to differentiate GWI before and after exercise in addition to between subgroups. We added the following line in the introduction of the paper for clarification:

Our group previously found that a submaximal exercise stress test was able to uncover neurological differences in the fMRI data of the Gulf War Illness subgroups (START: Stress Test Activated Reversible Tachycardia) and (STOPP: Stress Test Originated Phantom Perception) while performing a test of attention (the N-back working memory task). 

Using this same protocol we also demonstrated that Chronic Fatigue Syndrome (CFS) fMRI data was differentiable from a sedentary control using a multivariate pattern of activation and machine learning (a logistic regression algorithm). As a result we hypothesized that a machine learning algorithm such as a logistic regression estimation method used on the fMRI data of GWI subjects after exercise would potentially identify multivariate patterns of brain activation to distinguish GWI and its sub-groups from control subjects as well.

The following citations support this finding from our group:

  1. Rayhan RU, Stevens BW, Raksit MP, Ripple JA, Timbol CR, Adewuyi O, VanMeter JW, Baraniuk JN. Exercise challenge in Gulf War Illness reveals two subgroups with altered brain structure and function. PLoS One 2013;8:e63903. pmid:23798990
  2. Rayhan RU, Stevens BW, Timbol CR, Adewuyi O, Walitt B, VanMeter JW, et al. Increased brain white matter axial diffusivity associated with fatigue, pain and hyperalgesia in Gulf War illness. PLoS One 2013;8:e58493. pmid:23526988
  3. Baraniuk JN, El-Amin S, Corey R, Rayhan R, Timbol C. Carnosine treatment for gulf war illness: a randomized controlled trial. Glob J Health Sci 2013;5:69–81. pmid:23618477
  4. Clarke T, Jamieson J, Malone P, Rayhan R, Washington S, VanMeter J, Baraniuk J. Connectivity differences between Gulf War Illness (GWI) phenotypes during a test of attention. PLOS One. December 31, 2019. https://doi.org/10.1371/journal.pone.0226481
  5. Rayhan RU, Stevens BW, Raksit MP, et al. Exercise Challenge in Gulf War Illness Reveals Two Subgroups with Altered Brain Structure and Function. Valdes-Sosa PA, ed. PLoS ONE. 2013;8(6):e63903. doi:10.1371/journal.pone.0063903.
  6. Garner RS, Rayhan RU, Baraniuk JN. Verification of exercise-induced transient postural tachycardia phenotype in Gulf War Illness. Am J Transl Res. 2018 Oct 15;10(10):3254-3264. eCollection 2018. PubMed PMID: 30416666; PubMed Central PMCID: PMC6220213.
  7. Stuart D Washington, Rakib U Rayhan, Richard Garner, Destie Provenzano, Kristina Zajur, Florencia Martinez Addiego, John W VanMeter, James N Baraniuk, Exercise alters cerebellar and cortical activity related to working memory in phenotypes of Gulf War Illness, Brain Communications, Volume 2, Issue 1, 2020, fcz039, https://doi.org/10.1093/braincomms/fcz039
  8. Stuart D Washington, Rakib U Rayhan, Richard Garner, Destie Provenzano, Kristina Zajur, Florencia Martinez Addiego, John W VanMeter, James N Baraniuk, Exercise alters brain activation in Gulf War Illness and Myalgic Encephalomyelitis /

Chronic Fatigue Syndrome, Brain Communications, in press

The updated manuscript including edits in red is attached here. 

The authors would like to conclude this by thanking you very much so for your time and attention while conducting this review. We understand how much effort is involved in reading a paper and putting out such thoughtful insights and we really appreciate it.

Reviewer 2 Report

This study represents an attempt to identify patterns of brain activity that distinguish Gulf War Illness (GWI) individuals from normal controls who have a low level of physical activity with and without a physical exercise intervention. There are a number of serious questions about this work. The authors indicate that GWI represents a collection of vague symptoms without a clear pathophysiology.

The justification of using an exercise manipulation is not clearly justified and subsequently the choice of a sedentary control group is questionable.

The assumption that an N-back task during scanning has some clear physiological foundation and appropriate test-re-test properties is highly questionable. There is no indication that N-back performance is effected by exercise.

There are no data on task performance during testing for the groups under each exercise condition making it impossible to even guess how performance differed under each condition.

The foundation of the data processing is "activation" which significantly curtails the data subjected to subsequent analysis.

The data analysis is purely algorithmic without any physiological focus.

In spite of the large number of regions identified as discriminators, Figure 4 suggests that the GWI data are noisy (poorly structured) and not well replicated on day 2. Figure 5 suggests that this was especially true for the START and STOPP subgroups.

In spite of the large amount of computational effort, the results fail to reveal anything about GWI or exercise.

Author Response

Review 2:

  1. This study represents an attempt to identify patterns of brain activity that distinguish Gulf War Illness (GWI) individuals from normal controls who have a low level of physical activity with and without a physical exercise intervention. There are a number of serious questions about this work. The authors indicate that GWI represents a collection of vague symptoms without a clear pathophysiology.

The authors appreciate that the reviewer acknowledges the point of this study. The authors also hope with revisions that they can convince the reviewer that these questions have been rectified. GWI is indeed often interpreted as a collection of vague symptoms without a clear pathophysiology, despites the subjects experiencing it reporting very real pain and fatigue. Symptoms were consolidated by Steele (2000) by odds ratios of deployed vs. nondeployed 1990-1991 veterans in Kansas. The robust nature of this symptoms set has been demonstrated in other studies. (Maule 2017). Other studies and meta analysis (Maule 2017) We required moderate or severe levels of symptoms and did not allow mild or trivial symptoms (Fappiano 2020). The sense of “vagueness” is due to the need for a better understanding of brain perceptions of nociceptive, interoceptive, and autonomic afferent information and new framework for disease conceptualization. This analysis is one step in the larger effort to objectively define brain diseases in a similar fashion to the NIH RDoC philosophy. We feel it is especially important to demonstrate that it can be categorized accordingly.

  1. The justification of using an exercise manipulation is not clearly justified and subsequently the choice of a sedentary control group is questionable.

The authors appreciate the reviewer’s indication of a need to explain why we chose to use an exercise based task during a test of working memory. Our group previously found that the GWI subgroups were differentiable using fMRI data after the bicycle stress test. As such we had hypothesized the machine learning model would uncover a similar change. What we found instead was that using machine learning was able to differentiate GWI before and after exercise in addition to between subgroups. We added the following line in the introduction of the paper for clarification:

Our group previously found that a submaximal exercise stress test was able to uncover neurological differences in the fMRI data of the Gulf War Illness subgroups (START: Stress Test Activated Reversible Tachycardia) and (STOPP: Stress Test Originated Phantom Perception) while performing a test of attention (the N-back working memory task). 

Using this same protocol we also demonstrated that Chronic Fatigue Syndrome (CFS) fMRI data was differentiable from a sedentary control using a multivariate pattern of activation and machine learning (a logistic regression algorithm). As a result we hypothesized that a machine learning algorithm such as a logistic regression estimation method used on the fMRI data of GWI subjects after exercise would potentially identify multivariate patterns of brain activation to distinguish GWI and its sub-groups from control subjects as well.

Citation: Provenzano D, Washington SD and Baraniuk JN (2020) A Machine Learning Approach to the Differentiation of Functional Magnetic Resonance Imaging Data of Chronic Fatigue Syndrome (CFS) From a Sedentary Control. Front. Comput. Neurosci. 14:2. Doi: 10.3389/fncom.2020.00002

  1. The assumption that an N-back task during scanning has some clear physiological foundation and appropriate test-re-test properties is highly questionable. There is no indication that N-back performance is effected by exercise.

A series of prior papers from our group demonstrated that there were indications for neurological differences in GWI for the orthostatic subgroups START, STOPP, and POTS after exercise. Although not reported in this paper, details on the N-back performance are available in prior online supplementary material. Our introduction was updated to explain this and prior papers are cited here for review.

Intro:

Our group previously found that a submaximal exercise stress test was able to uncover neurological differences in the fMRI data of the Gulf War Illness subgroups (START: Stress Test Activated Reversible Tachycardia) and (STOPP: Stress Test Originated Phantom Perception) while performing a test of attention (the N-back working memory task). 

Citation:

  1. Rayhan RU, Stevens BW, Raksit MP, Ripple JA, Timbol CR, Adewuyi O, VanMeter JW, Baraniuk JN. Exercise challenge in Gulf War Illness reveals two subgroups with altered brain structure and function. PLoS One 2013;8:e63903. pmid:23798990
  2. Rayhan RU, Stevens BW, Timbol CR, Adewuyi O, Walitt B, VanMeter JW, et al. Increased brain white matter axial diffusivity associated with fatigue, pain and hyperalgesia in Gulf War illness. PLoS One 2013;8:e58493. pmid:23526988
  3. Baraniuk JN, El-Amin S, Corey R, Rayhan R, Timbol C. Carnosine treatment for gulf war illness: a randomized controlled trial. Glob J Health Sci 2013;5:69–81. pmid:23618477
  4. Clarke T, Jamieson J, Malone P, Rayhan R, Washington S, VanMeter J, Baraniuk J. Connectivity differences between Gulf War Illness (GWI) phenotypes during a test of attention. PLOS One. December 31, 2019. https://doi.org/10.1371/journal.pone.0226481
  5. Rayhan RU, Stevens BW, Raksit MP, et al. Exercise Challenge in Gulf War Illness Reveals Two Subgroups with Altered Brain Structure and Function. Valdes-Sosa PA, ed. PLoS ONE. 2013;8(6):e63903. doi:10.1371/journal.pone.0063903.
  6. Garner RS, Rayhan RU, Baraniuk JN. Verification of exercise-induced transient postural tachycardia phenotype in Gulf War Illness. Am J Transl Res. 2018 Oct 15;10(10):3254-3264. eCollection 2018. PubMed PMID: 30416666; PubMed Central PMCID: PMC6220213.
  7. Stuart D Washington, Rakib U Rayhan, Richard Garner, Destie Provenzano, Kristina Zajur, Florencia Martinez Addiego, John W VanMeter, James N Baraniuk, Exercise alters cerebellar and cortical activity related to working memory in phenotypes of Gulf War Illness, Brain Communications, Volume 2, Issue 1, 2020, fcz039, https://doi.org/10.1093/braincomms/fcz039
  8. Stuart D Washington, Rakib U Rayhan, Richard Garner, Destie Provenzano, Kristina Zajur, Florencia Martinez Addiego, John W VanMeter, James N Baraniuk, Exercise alters brain activation in Gulf War Illness and Myalgic Encephalomyelitis /

Chronic Fatigue Syndrome, Brain Communications, in press

  1. There are no data on task performance during testing for the groups under each exercise condition making it impossible to even guess how performance differed under each condition.

The authors appreciate the reviewer’s insight into the need for additional screening, demographic, and performance data. Subject data was detailed more extensively in prior publications and we have updated the methods section as such to reflect this. We required moderate or severe levels of symptoms and did not allow mild or trivial symptoms (Fappiano 2020). In regards to screening for hearing or vision skills and the performance on the N-back working task (SC performed better than GWI) a prior study published the results of the test, however this could be included as a supplement in this paper as well if the reviewer so desires. The methods section was updated with the following:

Further information regarding the study protocol, screening, demographics, subject symptoms, subject pain perception, orthostatic measurements, interoceptive complaints, chemical sensitivity questionnaires, and subject quality of life domain data are reported in previous published articles from our group and online as supplementary materials.    It should be noted that all subjects were screened for ability to perform the task prior to fMRI data collection, and were able to practice the N-back memory task until they felt comfortable prior to recording. Performance data from the GWI group was reported in a previous study to be markedly lower (on average about 26% lower) than the sedentary control both before and after exercise.

A prior study published the results of the test, however this could be included as a supplement in this paper as well if the reviewer so desires.

  1. The foundation of the data processing is "activation" which significantly curtails the data subjected to subsequent analysis.

This is true and the analysis was curtailed. Prior papers have detailed some of the additional details surrounding this clinical trial and are available here:

 Rayhan RU, Stevens BW, Raksit MP, Ripple JA, Timbol CR, Adewuyi O, VanMeter JW, Baraniuk JN. Exercise challenge in Gulf War Illness reveals two subgroups with altered brain structure and function. PLoS One 2013;8:e63903. pmid:23798990

 Rayhan RU, Stevens BW, Timbol CR, Adewuyi O, Walitt B, VanMeter JW, et al. Increased brain white matter axial diffusivity associated with fatigue, pain and hyperalgesia in Gulf War illness. PLoS One 2013;8:e58493. pmid:23526988

 Baraniuk JN, El-Amin S, Corey R, Rayhan R, Timbol C. Carnosine treatment for gulf war illness: a randomized controlled trial. Glob J Health Sci 2013;5:69–81. pmid:23618477

 Clarke T, Jamieson J, Malone P, Rayhan R, Washington S, VanMeter J, Baraniuk J. Connectivity differences between Gulf War Illness (GWI) phenotypes during a test of attention. PLOS One. December 31, 2019. https://doi.org/10.1371/journal.pone.0226481

  Rayhan RU, Stevens BW, Raksit MP, et al. Exercise Challenge in Gulf War Illness Reveals Two Subgroups with Altered Brain Structure and Function. Valdes-Sosa PA, ed. PLoS ONE. 2013;8(6):e63903. doi:10.1371/journal.pone.0063903.

 Garner RS, Rayhan RU, Baraniuk JN. Verification of exercise-induced transient postural tachycardia phenotype in Gulf War Illness. Am J Transl Res. 2018 Oct 15;10(10):3254-3264. eCollection 2018. PubMed PMID: 30416666; PubMed Central PMCID: PMC6220213.

Stuart D Washington, Rakib U Rayhan, Richard Garner, Destie Provenzano, Kristina Zajur, Florencia Martinez Addiego, John W VanMeter, James N Baraniuk, Exercise alters cerebellar and cortical activity related to working memory in phenotypes of Gulf War Illness, Brain Communications, Volume 2, Issue 1, 2020, fcz039, https://doi.org/10.1093/braincomms/fcz039 

Stuart D Washington, Rakib U Rayhan, Richard Garner, Destie Provenzano, Kristina Zajur, Florencia Martinez Addiego, John W VanMeter, James N Baraniuk, Exercise alters brain activation in Gulf War Illness and Myalgic Encephalomyelitis /

Chronic Fatigue Syndrome, Brain Communications, in press

  1. The data analysis is purely algorithmic without any physiological focus.

The authors appreciate this comment. For our paper we adopted a strict algorithmic approach as the focus was on machine learning. To tease out how an algorithm could predict physiological results our group chose to use a logistic regression to evaluate if GWI could be predicted from an SC and if the three previously demonstrated Orthostatic phenotypes found in GWI could be distinguished. Although the analysis is not physiological by nature, it shows how one could use algorithms to predict physiology and symptoms.

  1. In spite of the large number of regions identified as discriminators, Figure 4 suggests that the GWI data are noisy (poorly structured) and not well replicated on day 2. Figure 5 suggests that this was especially true for the START and STOPP subgroups.

This is consistent with our findings. Simple analysis using traditional means did not find easily identifiable differences in activation between GWI and an SC. One of the reasons we opted to pursue a machine learning approach was to test if it could tease out a pattern in the four groups. The differences after exercise demonstrated that GWI showed different activations after exercise.

If anything the noisy data suggests the strength in using machine learning and a multivariate pattern to attempt to distinguish the disorders. Distinguishing GWI is a hard problem that carries a lot of controversy, but the symptoms experienced by subjects are very real and if a multivariate pattern can help distinguish the disorder, it could potentially assist subjects to find appropriate remediation in the future.

  1. In spite of the large amount of computational effort, the results fail to reveal anything about GWI or exercise.

The goal of this paper was not to tease out the effects of exercise on the brain or physiological underpinnings of GWI for any one region. Instead the goal was to reveal how a multivariate pattern of brain activations could be used to distinguish GWI from a sedentary control. Prior studies regarding the n-back task and exercise (as cited above) found that differences in the orthostatic subgroups were identifiable for GWI subjects after exercise. This study showed that despite the lack of an underlying uniform pathophysiology, GWI can be differentiated from an SC.

Despite there being data regarding exercise affects for sedentary control individuals as well while undergoing a test of attention, we did not focus on the affects of exercise on fMRI data during exercise.

This paper reveals how one could apply a previously published methodology to characterize and potentially diagnose a hard to diagnose disorder using machine learning to GWI, and then successfully is able to differentiate it. The ultimate multivariate pattern does indicate some of these regions are important at predicting GWI, and therefore might be important for GWI, however it is difficult from this analysis to pinpoint any one region, and isn’t a focus of it. We feel this analysis has merit as it paves the way for further incorporation of machine learning into medical imaging, creation of new biomarkers and diagnostic tools using machine learning, and assists GWI veterans with further peace of mind that they do in fact experience a uniform condition that can be explained by differing multivariate patterns of brain activity.

The attached manuscript has all updates detailed in red text. 

The authors would like to conclude this by thanking you very much so for your time and attention while conducting this review. We understand how much effort is involved in reading a paper and putting out such thoughtful insights and we really appreciate it.

Round 2

Reviewer 1 Report

The authors have addressed the issues I raised satisfactorily in this revision.